https://doi.org/10.1038/s41467-022-31096-8　　OPEN

# Enhancement of electrocatalytic oxygen evolution by chiral molecular functionalization of hybrid 2D electrodes

Yunchang Liang [1,2✉], Karla Banjac [1,2], Kévin Martin[3], Nicolas Zigon [3], Seunghwa Lee[4,8], Nicolas Vanthuyne [5], Felipe Andrés Garcés-Pineda[6], José R. Galán-Mascarós [6,7], Xile Hu [4], Narcis Avarvari [3✉] & Magalí Lingenfelder [1,2✉]

A sustainable future requires highly efficient energy conversion and storage processes, where electrocatalysis plays a crucial role. The activity of an electrocatalyst is governed by the binding energy towards the reaction intermediates, while the scaling relationships prevent the improvement of a catalytic system over its volcano-plot limits. To overcome these limitations, unconventional methods that are not fully determined by the surface binding energy can be helpful. Here, we use organic chiral molecules, i.e., hetero-helicenes such as thiadiazole-[7] helicene and bis(thiadiazole)-[8]helicene, to boost the oxygen evolution reaction (OER) by up to ca. 130 % (at the potential of 1.65 V vs. RHE) at state-of-the-art 2D Ni- and NiFe-based catalysts via a spin-polarization mechanism. Our results show that chiral molecule-functionalization is able to increase the OER activity of catalysts beyond the volcano limits. A guideline for optimizing the catalytic activity via chiral molecular functionalization of hybrid 2D electrodes is given.

[1] Max Planck-EPFL Laboratory for Molecular Nanoscience and Technology, École Polytechnique Fédérale de Lausanne (EPFL), 1015 Lausanne, Switzerland. [2] Institute of Physics (IPHYS), École Polytechnique Fédérale de Lausanne (EPFL), 1015 Lausanne, Switzerland. [3] Univ Angers, CNRS, MOLTECH-Anjou, SFR MATRIX, F-49000 Angers, France. [4] Laboratory of Inorganic Synthesis and Catalysis, Institute of Chemical Sciences and Engineering, École Polytechnique Fédérale de Lausanne (EPFL), 1015 Lausanne, Switzerland. [5] Aix Marseille Université, CNRS, Centrale Marseille, iSm2, Marseille, France. [6] Institute of Chemical Research of Catalonia (ICIQ), The Barcelona Institute of Science and Technology (BIST), Av. Països Catalans 16, E-43007 Tarragona, Spain. [7] Catalan Institution for Research and Advanced Studies (ICREA), Passeig Lluis Com- panys, 23, Barcelona 08010, Spain. [8]Present address: Department of Chemical Engineering, Changwon National University, 51140 Changwon, South Korea. ✉email: yunchang.liang@epfl.ch; narcis.avarvari@univ-angers.fr; magali.lingenfelder@epfl.ch

Efficient electrocatalysis, especially electrocatalytic water-splitting, is imperative in future sustainable energy systems and carbon neutrality processes[1,2]. However, the complex reaction pathways and sluggish kinetics of the oxygen evolution reaction (OER) impede the application of electrocatalytic water-splitting[3]. The Sabatier principle states that an ideal catalyst surface should bind to the reaction intermediates with optimal strength (neither too weak nor too strong)[4]. Moreover, the binding energies of the variables of one adsorbate species to a catalyst surface follow a scaling relationship with each other. For instance, the adsorption free energies of OER reaction intermediates OH* and OOH* adhere to a simple relation, $\Delta G_{OOH*} = \Delta G_{OH*} + 3.2 \pm 0.2$ eV, for a large number of catalysts[5,6]. Consequently, a volcano-type plot confines the activity of the catalysts and guides the development of new catalytic surfaces[7–9]. Therefore, conventional strategies to enhance the catalytic activity are mostly based on adjusting the binding energy. Unconventional approaches, including magnetic control, can improve the electrocatalysis efficiency beyond the volcano limitations and are receiving increasing attention[10–12].

Electron transfer takes place during electrocatalysis processes. One of the inherent properties of electrons is the electron spin. Recently, it was reported that electron transport through a chiral molecule induces a spin polarization without external magnetic fields, i.e., after passing the molecule an imbalance between up and down spin is created[13–16]. This so-called chiral-induced spin selectivity (CISS) effect was suggested applicable in spin-dependent electron transfer processes at liquid/solid interfaces, for instance, the electrocatalytic OER[8,17]. It was suggested that the CISS process affects the spin of the electrons at the catalyst surface, consequently influencing the reaction pathway and enhancing the oxygen evolution. Therefore, it provides a unique opportunity to improve the activity and selectivity of the OER beyond the volcano plot-related restrictions.

However, experimentally decoupling the various coexisting effects contributing to OER is essential to assess the effect of chiral molecular functionalization. In particular, the high dissolution rate of catalyst materials (e.g., Fe) under OER conditions[18], the weak bonding of thiol groups on metal oxides[19,20], the influence of organic molecules and their different structural conformers on the electrochemically active surface area (ECSA) and the coordination environment of the active surface sites[21] need to be considered. Therefore, well-defined systems are essential to decouple the effects of chiral molecular functionalization on the OER activity. Here, we use hybrid electrodes composed of a monolayer or sub-monolayer of rigid chiral (achiral) molecules and 2D electrocatalysts in a sandwich configuration.

2D Ni (NiO$_x$) and NiFe (NiFeO$_x$) oxo-hydroxides are known as state-of-the-art electrocatalysts in alkaline media[22]. Carbohelicenes contain *ortho*-condensed aromatic rings with helical structures that possess axial chirality, and 2D self-assembly of helicene molecules on metals bestow chiral properties to the surface[23]. Their high racemization barriers[24] preserve their enantiopurity at the liquid/solid interface during the OER. The rigid structure of the helicenes also limits the number of molecular conformers at the electrode surface[25–28]. However, carbohelicenes interact weakly with metallic substrates (for instance, Cu, Ag, and Au) via van der Waals forces, resulting in high mobility of the molecules at room temperature[19,29]. Attachment of additional functional groups on helicenes is thus needed to increase the stability of such a hybrid electrode by strengthening the bonding to the metal surface[30].

Here, using chiral fused thiadiazole-helicene molecules, we report on the role of spin polarization by chiral molecular functionalization on the OER activity of metallic electrodes and the optimization of the chiral molecule-catalyst configuration for optimal activity.

## Results and discussion

**Effect of the molecular functionalization on the OER activity.** In the first set of experiments, enantiopure (*M*)- and (*P*)-thiadiazole-[7]helicenes, previously reported by some of us[31] in the frame of our general interest in chiral molecular materials[32], have been used for the chiral molecular functionalization of metal electrodes.

Figure 1 shows the effect of the adsorbed thiadiazole-[7] helicene enantiomers on the OER activity of two types of electrodes, namely Au(111) with monolayer NiO$_x$ islands and bare Au(111), in O$_2$-saturated 0.1 M aqueous KOH solution. Farhat et al.[33] reported that the OER activity of ultrathin NiO$_x$ and NiFeO$_x$ films under reaction conditions decreases over time in Fe-free KOH but stays more constant in unpurified (i.e., containing trace Fe impurities) KOH after Fe incorporation. Therefore, an unpurified KOH solution has been used to ensure all activity measurements are reproducible.

A clear enhancement in the OER current can be seen after the helicene molecules were deposited on NiO$_x$ samples, as shown in Fig. 1a, b. Deposition of (*P*)- and (*M*)-thiadiazole-[7]helicene enhances the OER current at 1.65 V vs. RHE ~85% and ~74%, respectively. Electrochemical impedance spectroscopy (EIS) measurements were conducted before and after the molecule deposition to determine if the current enhancement could be just due to an increment in the ECSA during the molecule deposition process. The ECSA (proportional to the active surface) was obtained by determining the adsorption capacitance around the "onset" potential of the OER[34] (Supplementary Fig. 1a). The OER currents normalized by the ECSA values still uphold the enhancement effect, i.e., ca. 61% increase in the current density at 1.65 V vs. RHE and ca. 33.4 mV reduction in the overpotential at the current density of 10 mA cm$^{-2}$ in the case of (*P*)-thiadiazole-[7]helicene (Supplementary Fig. 1b). It is likely that the increase in the adsorption capacitance in Supplementary Fig. 1a was mainly induced by the adsorbed helicene molecules[35,36], and not by the increment of the ECSA. A comparison of activity normalized by the EIS-determined ECSA and the Ni redox peaks is shown in Supplementary Fig. 1c. Nevertheless, the normalized current reinforces the conclusion that the thiadiazole-[7]helicene enantiomers are able to enhance the OER at NiO$_x$ islands on Au surfaces.

Although the effectiveness of the helicene molecules on OER enhancement at NiO$_x$ is validated, it is likely that molecules were not directly adsorbed on the NiO$_x$ islands as no metallic Ni sites were accessible during the measurements. Direct thiadiazole bonding on metal oxide surfaces is difficult under the current conditions[37]. Therefore, the molecules were mostly adsorbed on the Au substrate. To test the effect of direct bonding between the helicene molecule and the catalyst surface on the OER, bare Au(111) electrodes were used for analogous OER experiments (Fig. 1c). Although such chirally functionalized Au(111) has lower OER activity compared to NiO$_x$, we anticipate that the activity measurements on Au(111) help to elucidate the effect of direct adsorption of helicene molecules at OER active sites. In contrast to the results from NiO$_x$ samples, the presence of (*P*)-thiadiazole-[7]helicene molecules reduced the OER at the Au(111) surface. This difference can be ascribed to the different catalyst-helicene molecule configurations. At the NiO$_x$ decorated Au substrates, the molecules were mostly adsorbed on the Au surfaces, and the active centers at NiO$_x$ islands were not directly affected. At bare Au samples, the OER was taking place at the Au surface, which was partially blocked by the molecules after deposition. The exposed surface area of the Au samples was determined by the stripping of Pb atoms deposited by the underpotential deposition (UPD) method[38] (see Supplementary Fig. 4a, b for the Pb stripping results and the normalized current

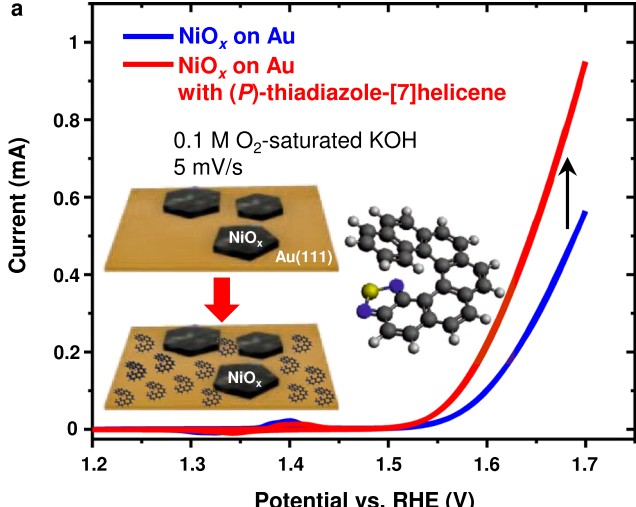

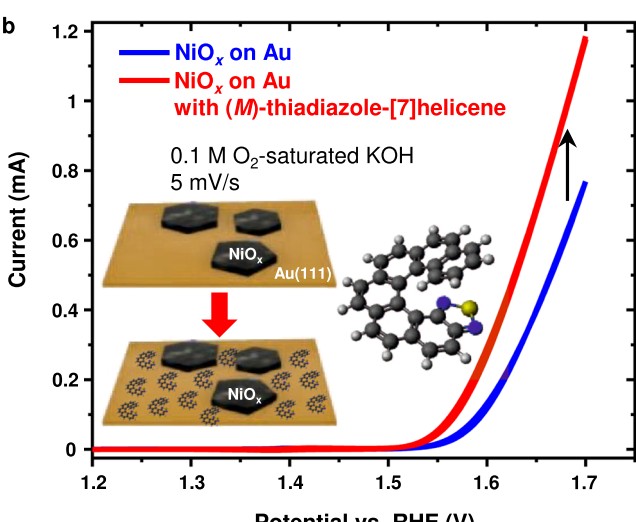

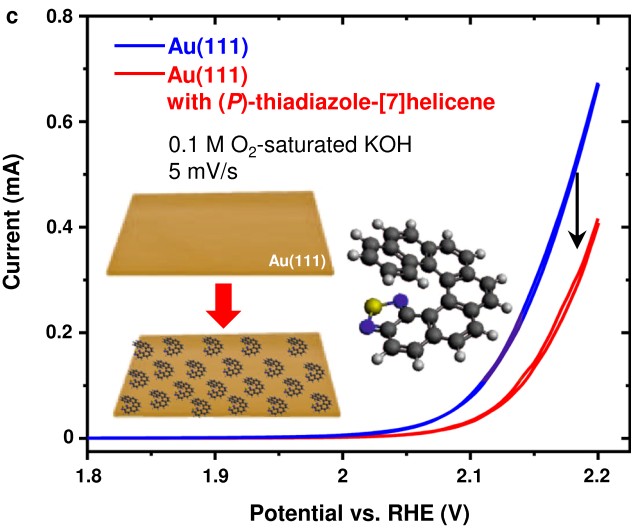

**Fig. 1 Helicene enantiomer functionalization effect on the OER activity.** OER activity of **a** (*P*)-thiadiazole-[7]helicene embedded NiO$_x$ electrodes, **b** (*M*)-thiadiazole-[7]helicene embedded NiO$_x$ electrodes, and **c** (*P*)-thiadiazole-[7]helicene functionalized Au(111) in 0.1 M O$_2$-saturated KOH. Source data are provided as a Source Data file.

density). The results show that a submonolayer of helicene molecules was on the sample surface after the deposition. The decrease in the OER current is not proportional to the decrease in the surface area of Au. The helicene molecules even reduced the specific current density. As shown in Supplementary Fig. 4a, peak 1, which can be assigned to low-coordination and step sites[38,39], decreased and shifted towards more negative potentials. Zwaschka et al.[40] found that OER activity on a polycrystalline Au surface is dominated by <1% of the surface consisting of defects. Therefore, the molecules were more stably adsorbed on surface defects during the reactions.

**Molecular functionalization vs. Fe-doping effect on OER**. Trotochaud et al.[41] studied the effect of Fe impurities in the electrolyte and Fe doping in the catalysts on the OER activity, showing that Fe doping is efficient to improve the OER activity. However, this improvement still adheres to the volcano-plot limits[42,43]. In this work, Fe impurities from the unpurified electrolyte had been incorporated in the NiO$_x$ samples during activity measurements before chiral molecular functionalization. In addition, we have also conducted OER activity measurements in Fe-free KOH. The Fe impurities in KOH were removed by precipitated bulk Ni(OH)$_2$[41]. Samples consisting of monolayer NiFeO$_x$ islands on Au surfaces were also used. A comparison of the activity enhancement caused by the (*P*)-thiadiazole-[7]helicene and the Fe doping in Ni-based catalysts is shown in Fig. 2. The Ni$_9$FeO$_x$ catalyst shown in Fig. 2b was deposited on a Ni foam using a combustion method[44]. The ECSA was obtained from the determination of the adsorption capacitance[34]. Figure 2a clearly shows that the helicene molecules increased the OER activity of Ni-based 2D islands modified by Fe doped directly in the catalysts or incorporated from the Fe-impurities-containing (unpurified) electrolyte. (*P*)-thiadiazole-[7]helicene reduces the overpotential at the current density of 1 mA cm$^{-2}$ by ~0.025, ~0.020, and ~0.023 V at NiO$_x$ islands in Fe-free KOH, NiO$_x$ islands in unpurified KOH and NiFeO$_x$ islands in unpurified KOH, respectively. The enhancement was higher on 2D NiFeO$_x$ islands deposited on a flat Au substrate than on Ni$_9$FeO$_x$ deposited on a Ni foam, i.e., ca. 120% and ca. 56%, respectively, at the potential of 1.55 V vs. RHE. It is presumably caused by the different amounts of helicene molecules that can be embedded into different electrodes. On the Au substrates, helicene molecules were stably bonded to the Au surface. However, the Ni foam electrode surface only consists of metal oxides, which do not form strong interactions with the helicene molecules. The Ni foam electrodes have much higher roughness compared to the Au electrodes. The helicene molecules were probably inserted into the porous structures of the Ni foam. Nevertheless, the effect caused by the chiral molecules is independent and compatible with the Fe-doping effect.

To confirm the enhancement is owing to the chirality of the adsorbed helicene molecules instead of other neglected properties of thiadiazole-[7]helicene and, at the same time, to investigate the influence of the helicene length on the OER, we synthesized bis(thiadiazole)-[8]helicene. The additional thiadiazole ring at the opposite end of the helix is expected to stabilize the Au-helicene-NiO$_x$ sandwich structure. We produced the racemic form of bis(thiadiazole)-[8]helicene and resolved the (*P*) and (*M*) enantiomers by chiral HPLC on Chiralpak IF (see "Methods" and the Supplementary Information for the synthesis, separation procedures and crystal structures). Suitable single crystals for the X-ray diffraction analysis have been obtained for both the enantiopure and also the racemic helicenes. The first eluted enantiomer was the dextrorotatory one, having a specific optical

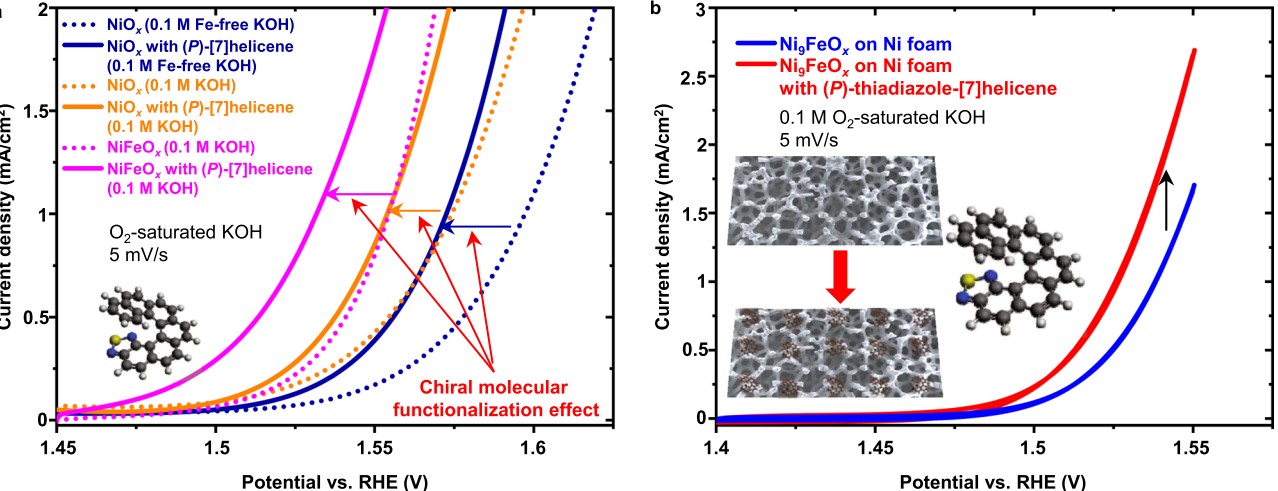

**Fig. 2 Comparison of chiral molecule effect with Fe-doping effect. a** Chiral molecular functionalization enhancement on $NiO_x$ islands in Fe-free, $NiO_x$ islands in KOH (unpurified) and $NiFeO_x$ islands in KOH (unpurified). **b** Chiral molecular functionalization enhancement at $Ni_9FeO_x$ deposited on Ni foam. Source data are provided as a Source Data file.

rotation of $[\alpha]_D^{25} = + (8100 \pm 1\%)°$, corresponding, as expected, to the clockwise (*P*) enantiomer, in agreement with the single-crystal X-ray analysis. Mirror-image circular dichroism (CD) spectra were obtained for the two enantiomers (see Supplementary Fig. 17). In the solid state, the helical curvature, characterizing the dihedral angle between the two terminal thiadiazole rings, amounts at 29° for the racemate and 35° for the enantiopure compounds, thus smaller than 45° observed for the thiadiazole-[7]helicenes[31]. Then, with the new bis(thiadiazole)-[8]helicene in our hands, we set out to the chiral molecular functionalization of our electrodes.

Figure 3a shows that the activity of $NiO_x$ on Au was improved by the presence of (*M*)-bis(thiadiazole)-[8]helicene molecules on the surface, with ca. 130% enhancement at the potential of 1.65 V vs. RHE. A statistical analysis of enhancement at five nominally identical $NiO_x$ electrodes can be seen in Supplementary Fig. 5. The (*M*)-bis(thiadiazole)-[8]helicene yields a sensibly greater improvement in the OER current than the thiadiazole-[7]helicenes. The much more intense OER activity enhancement by the bis(thiadiazole)-[8]helicene functionalized electrodes compared to the thiadiazole-[7]helicene functionalized ones is in favor of this sandwich-type structure of the electrodes, with the helicene lying between the Au substrate and the $NiO_x$ islands. It can be hypothesized that the former strongly interacts with both the substrate and the catalyst thanks to the presence of the functional thiadiazole rings on both sides of the helical connector.

So far, the results show that chiral helicene molecules can enhance the OER activity. However, the compound in the molecules that directly bonds to the electrode surface is the thiadiazole group. Thus, it is essential to evaluate the role of thiadiazole. Accordingly, we used 2,1,3-benzothiadiazole, a simple achiral molecule containing a thiadiazole cycle fused to a benzene ring. It has the same molecular footprint bonds to the Au(111) surface as the helicene molecules used in this work. As shown in Fig. 3b, the presence of the achiral molecules did not noticeably influence the OER activity of $NiO_x$ islands on Au. Therefore, the interaction between the thiadiazole compound and the electrode surface is not the origin of the OER enhancement activity.

To maximize the chiral molecular functionalization effect, we deposited the $NiO_x$ islands on Au surface covered by a monolayer of (*M*)-bis(thiadiazole)-[8]helicene. The activity of this sandwich configuration was compared to $NiO_x$ islands (same $NiO_x$ loading)

directly deposited on Au surface without helicene molecules. The results are shown in Fig. 3c. The enhancement in the current (ca. 250% at 1.65 V vs. RHE without iR correction) is likely greater than the case where helicene molecules were added to the electrodes after $NiO_x$ islands.

Water oxidation is a suitable example to test the chiral molecular functionalization effect on product selectivity, as $H_2O_2$ formation is expected to be reduced at spin-polarized surfaces[45]. However, $NiO_x$ is not among the catalysts (e.g., ZnO[46], $SnO_2$, and $TiO_2$[47]) that selectively produce $H_2O_2$ during water oxidation. The OER at $NiO_x$-based catalysts already dominates in alkaline media[48], and the production of $H_2O_2$ is neglectable in these conditions. Moreover, it is reported that a neutral solution containing bicarbonate enhances the faradaic efficiency towards $H_2O_2$ production[49], and the neutral pH also hinders the $H_2O_2$ from decomposition. In order to test for changes in selectivity induced by the chiral molecules, we used the same electrode configurations in Fig. 3c to confirm that the chiral molecular functionalization suppresses $H_2O_2$ production at neutral pH. Chronoamperometry (Supplementary Fig. 6) and UV-visible spectrophotometry measurements (see "Methods") were conducted in $CO_2$ saturated $K_2CO_3$ (neutral pH and containing a constant concentration of $HCO_3^-$). The $H_2O_2$ production was significantly reduced by the chiral molecular functionalization, as shown in Fig. 3d. This is because $H_2O_2$ (a singlet species) formation is spin forbidden at a spin-polarized surface. This finding is in line with the CISS effect in water-splitting reported previously[13,17,45].

On Au(111) surfaces, the helicene molecules formed self-assembled monolayer (SAM) structures. The STM images of (*P*)-thiadiazole-[7]helicene and (*M*)-bis(thiadiazole)-[8]helicene molecules on Au(111) are shown in Fig. 4a, b, respectively (see Supplementary Fig. 7 for larger-scale images showing different domains of assemblies). The former formed trimeric structures with ~0.75 molecules/$nm^2$, while the latter formed rows of dimers with ~1.1 molecules/$nm^2$. In the SAM of (*M*)-bis(thiadiazole)-[8] helicene molecules, the existence of three types of domains (60° to each other) is observed, as shown in Supplementary Fig. 7b. These types of molecular assemblies are common to other functionalized helicenes on Au(111)[25–27,29]. The molecular coverage of functionalized helicenes on the substrates determines the assembly, going from trimeric structures at low coverage to dimeric rows at high coverage[26–29]. Therefore, the (*M*)-bis(thiadiazole)-[8]helicene

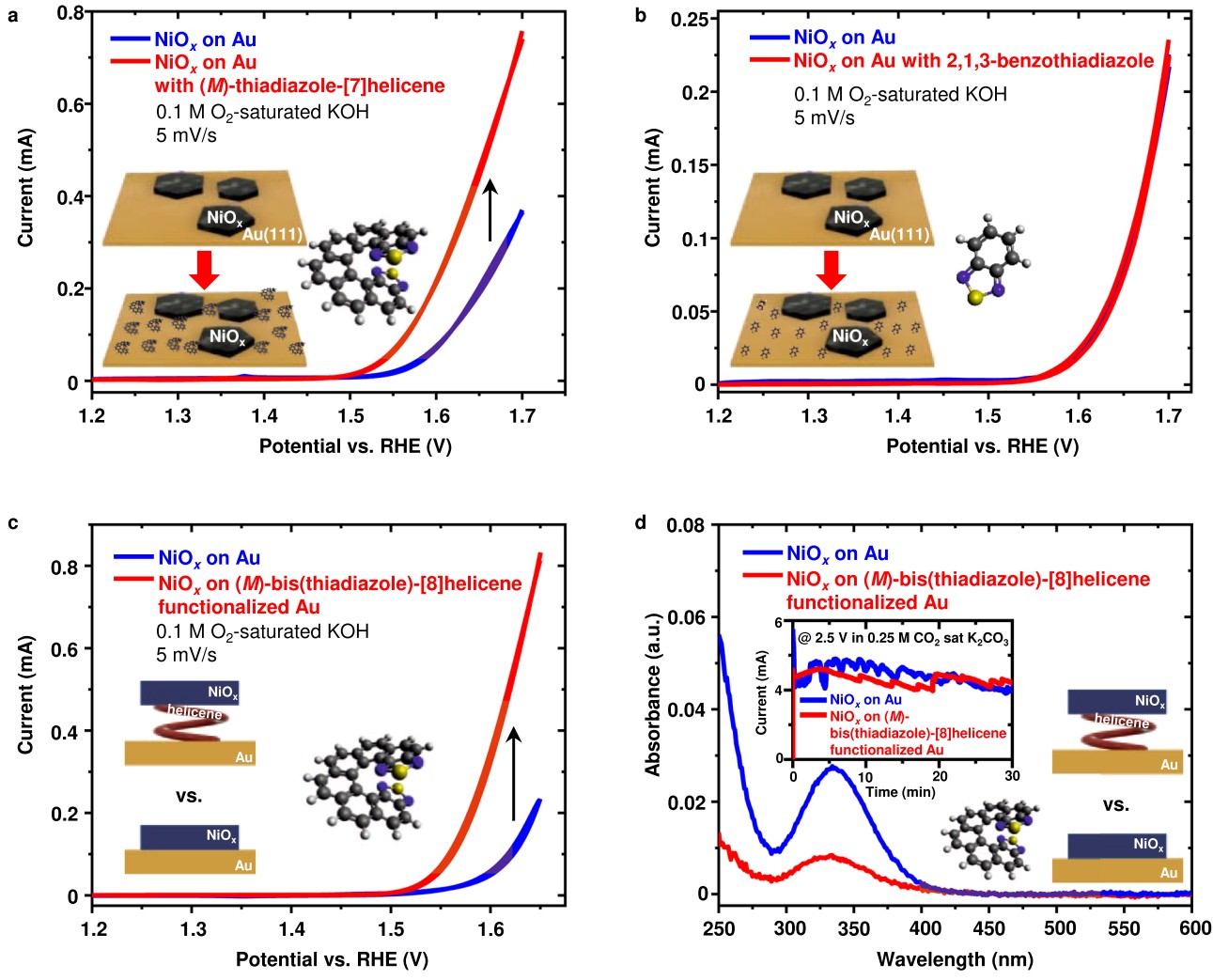

**Fig. 3 OER activity of NiO$_x$ with chiral and achiral molecules and chiral molecular functionalization effect on the product selectivity. a, b** OER activity of **a** (*M*)-bis(thiadiazole)-[8]helicene on Au surface with NiO$_x$, **b** 2,1,3-benzothiadiazole on Au with NiO$_x$ in 0.1 M O$_2$-saturated KOH. **c** Comparison of OER activity of NiO$_x$ islands deposited on Au and NiO$_x$ islands deposited on Au with a monolayer of (*M*)-bis(thiadiazole)-[8]helicene molecules. **d** UV-visible spectrophotometry analysis of H$_2$O$_2$ production from chronoamperometry at 2.5 V vs. RHE (inset, see more details in Supplementary Fig. 6) at NiO$_x$ islands on Au and NiO$_x$ islands on Au with a monolayer of (*M*)-bis(thiadiazole)-[8]helicene molecules. The absorbance spectra are normalized by the total charges transferred during the reaction. Source data are provided as a Source Data file.

reached a higher coverage than the (*P*)-thiadiazole-[7]helicene under similar conditions, likely due to the added thiadiazole group.

STM images were taken on the helicene molecule-functionalized NiO$_x$ samples after the OER measurements, as shown in Supplementary Fig. 8. The NiO$_x$ islands were roughened due to the OER, and the surfaces of the islands were likely not covered by the molecules. However, the structures on the Au surface suggest that the helicene molecule assembly on Au was stable during the OER. Bare Au surfaces can be seen in several areas in the images, which is in line with the Pb stripping results in Supplementary Fig. 4.

The activity enhancement using chiral molecules appears to depend on the anchoring of the molecules with regard to the catalysts. Based on our results, the dependence of the OER activity enhancement on the molecule-catalyst configuration is presented in Fig. 4c. Helicene molecules on top of the catalyst surface block the active centers and consequently reduce the overall catalytic activity. Therefore, electrode manufacturing is vital to the performance of helicene molecule-functionalized catalysts. A rationally designed electrode requires the catalytically active centers free of blockage and the electron transfer through

the spin polarizers. For an optimal activity enhancement, the chiral molecules should be between the catalytic material and the substrate. A substrate that can strongly bond the chiral molecules (for example, Au in this work) is preferred for fabricating a stable electrode.

**Chiral molecular functionalization.** In this work, we show that chiral molecular functionalization can improve the OER activity of metal oxide catalysts. A specific catalyst-chiral helicene molecule-Au substrate configuration (as shown in Fig. 4c) shows the optimal catalytic activity. By comparing the effect of chiral helicene molecules and the achiral 2,1,3-benzothiadiazole, it is evident that such activity enhancement is directly related to the chirality of the molecules, rather than ligand effects introduced by the bonding of the molecules on the catalyst surface. The bonding between the catalyst and the chiral molecule alone does not cause any improvement in the catalytic activity. Contrarily, the bonding can reduce the activity when the chiral molecules are directly deposited at the catalyst active sites. The results of this comprehensive study are in agreement with the interpretation from ref. [10] that a CISS-induced effect can efficiently enhance the OER.

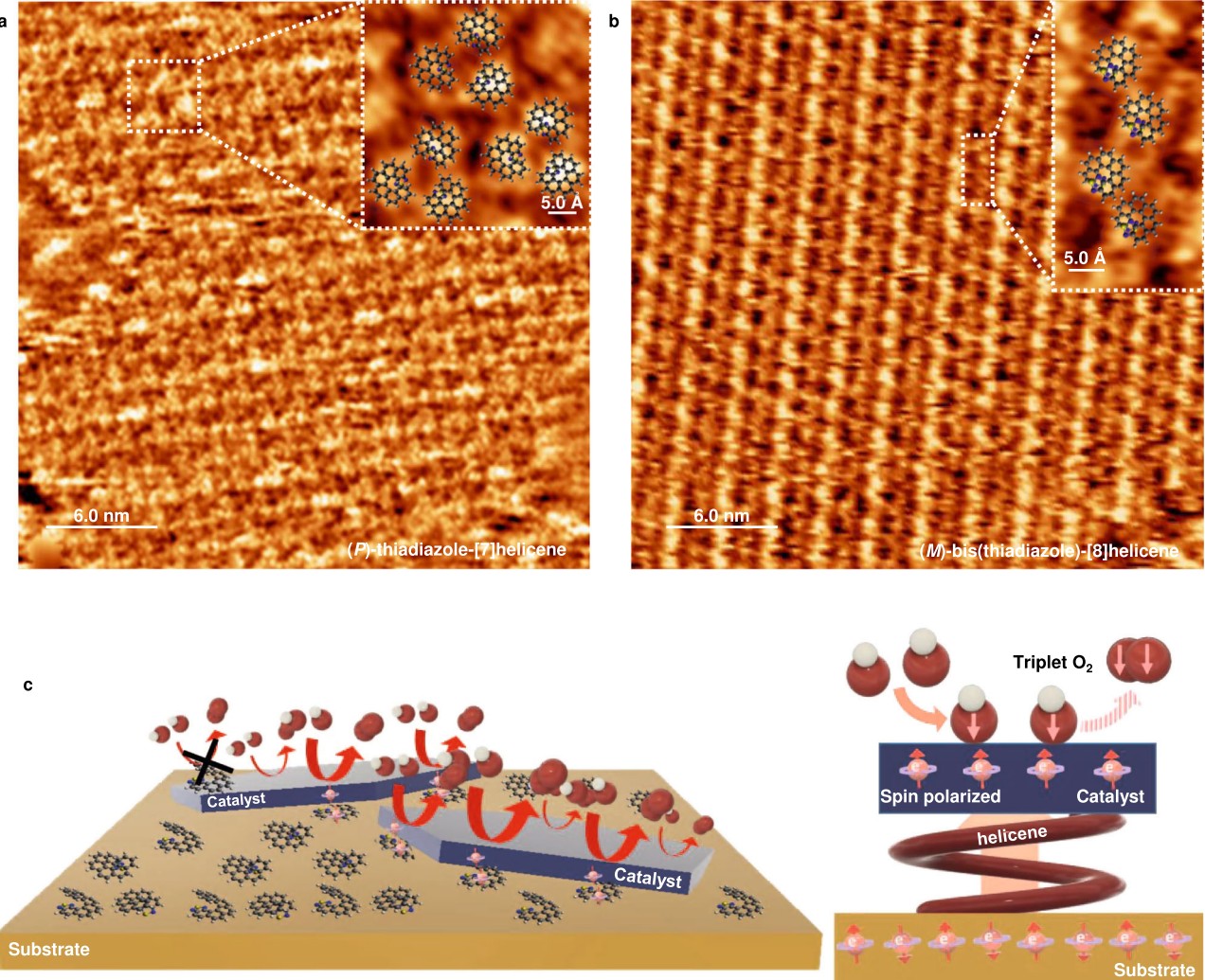

**Fig. 4 Helicene SAM and effect on the OER activity. a** An STM image of SAM of (*P*)-thiadiazole-[7]helicene (image size: 30 nm × 30 nm, tunneling current: 25 pA, sample bias: −5 mV). The dashed square marks three trimers of the molecules. A high-resolution image (image size: 4 nm × 4 nm, tunneling current: 20 pA, sample bias: 20 mV) showing three trimers is in the inset. **b** An STM image of SAM of (*M*)-bis(thiadiazole)-[8]helicene on Au(111) (image size: 30 nm × 30 nm, tunneling current: 20 pA, sample bias: 25 mV). The dashed rectangle marks two dimers of the molecules. The inset shows a high-resolution image (image size: 3 nm × 4 nm, tunneling current: 20 pA, sample bias: −25 mV) of two dimers. The 3D models of the molecules show the locations of the molecules. **c** Chiral molecule effect on the OER activity of NiO$_x$ islands dependent on the catalyst-chiral molecule-Au substrate configuration and illustration of the chiral molecular spin polarization effect on the OER.

In the CISS effect, the adsorption of chiral molecules on a surface spin-polarizes the interface[50–52]. In electron scattering, the spin polarization is induced by symmetry breaking[53]. This can be expanded to any process that can be described by a transition matrix, for example, photoemission, where the spin polarization is induced by the broken experimental symmetry also for light elements[54,55]. In electron transport, all contributions from the complex transition matrix elements are typically symmetric unless a chiral element is introduced. In this case, the transmitted electrons will acquire a spin polarization, similar to a paramagnetic material in an external magnetic field. Garcés-Pineda et al.[11] reported on the external magnetic field enhancement of the OER at various magnetic transition metal oxide-based catalysts. Although an activity enhancement can be caused by both chiral molecular functionalization and an external magnetic field, the enhancement magnitudes appear to be different. At the potential of 1.65 V vs. RHE, the OER current density (scan rate: 5 mV/s) on NiO$_x$-based catalysts in 0.1 M KOH increased ca. 61% by using (*P*)-thiadiazole-[7]helicene, which is significantly larger

than the ca. 10% in 1 M KOH (scan rate: 5 mV/s) caused by an external magnetic field (≤450 mT)[11]. Thus, chiral molecular functionalization may be comparatively more effective. It has also been well established that external magnetic fields can accelerate mass transfer processes through a magnetohydrodynamic effect[56–58]. The local magnetic field at the catalyst surface introduced by the chiral molecules and its effect on the local mass transfer should be further probed.

It was suggested that the electron spin at the catalyst surface strongly influences the bonding strength of the catalyst to oxygen species and the charge transfer between the catalyst and the oxygen adsorbates[11,59–61]. Electron spin polarization at catalytic surfaces with specifically modulated ferromagnetic properties leads to advanced OER kinetics[12,62–64], and a ferromagnetic catalyst can promote the spin polarization under an external magnetic field and thus enhances its OER activity[65]. However, not all practical OER catalysts are ferromagnetic. In addition, metal oxide-based catalysts, especially highly active 2D nanostructured catalysts, undergo severe surface reconstruction

processes under OER conditions[66–68]. The preservation of the well-prepared ferromagnetic properties under surface reconstruction is questionable. The chiral molecular functionalization provides a more versatile and sustainable electron spin polarization effect (Fig. 4c). It can be applied to most catalysts independent of their electronic (magnetic) properties and under different reaction conditions, i.e., it is less affected by the catalyst composition and surface reconstruction.

Compared to chiral molecular functionalization, Fe doping is a verified way to improve the OER activity of Ni-based catalysts[69]. It is revealed that the OER activity of Ni-based catalysts in non-specifically treated electrolytes (containing a trace amount of Fe) is affected by the Fe doping, and the catalysts effectively turn into NiFe dual catalysts[41]. Our results show that chiral molecular functionalization enhancement can coexist with the Fe-doping effect and other alloying methods since the helicene molecules do not interact directly with the metal oxide active centers. Therefore, it is compatible with most metal oxide-based catalysis systems and independent of the chemical composition of the catalysts.

Moreover, the newly introduced (M)-bis(thiadiazole)-[8]helicene brings a higher enhancement to the OER than (M)-thiadiazole-[7] helicene, i.e., ca. 130% vs. ca. 74% in the overall current at the potential of 1.65 V vs. RHE in 0.1 M KOH. This finding suggests the enhancement is related to the structure of the helicene molecules, for instance, molecular length and functional groups that determine the interactions with the catalyst and the substrates. Further enhancement through modification of the helicene molecules can be anticipated.

In summary, we have validated and evaluated the chiral molecular functionalization on the oxygen evolution at 2D hybrid chiral/achiral molecule-transition metal oxide electrodes. The results show that the chirality of the helicene molecules is accountable for a great enhancement in the activity of state-of-the-art OER catalysts. The enhancement is related to the electron spin polarization at the catalyst surface. However, the chiral molecular functionalization does not change the catalyst composition, and it is compatible with other enhancement methods, e.g., Fe doping. Therefore, this approach has the potential to boost electron spin-dependent catalytic reactions (e.g., the OER) beyond the common volcano-plot limits governed by the Sabatier principle and the scaling relationships. The comparison of different electrode configurations provides a clear guideline for optimizing the enhancement. This approach opens new avenues for developing next-generation catalytic systems with high efficiency and advanced selectivity through rational molecular functionalization.

## Methods

**Synthesis of (rac)-bis(thiadiazole)-[8]helicene.** For the reaction Scheme, full experimental details, characterization, chiral HPLC separation, and X-ray diffraction analysis details, see the Supplementary Information.

**2,7-bis(benzothiadiazol)vinyl)naphthalene 2.** 2,7-bis((triphenylphosphonium) methyl)naphthalene bromide **1** (0.26 g, 0.31 mmol, 1 equiv.) was suspended in 7 mL of dry THF under an argon atmosphere. The mixture was cooled down to −78 °C and then nBuLi (0.41 mL, 1.6 M in hexane, 0.65 mmol, 2.1 equiv.) was added dropwise. After 30 min at −78 °C, it was warmed to room temperature and stirred for one hour. The mixture was cooled down again to −78 °C and benzothiadiazole-5-carbaldehyde was added (0.16 g, 0.097 mmol, 3.1 equiv.). The mixture was stirred 10 min at −78 °C, then warmed to room temperature and left at RT overnight. After filtration on Celite® and concentration under vacuum, column chromatography (SiO₂, petroleum ether: CH₂Cl₂ 4: 6) yielded **2** as a yellow powder (90 mg, 45% yield, mixture of Z/E isomers). ¹H NMR (300 MHz, CDCl₃): δ (ppm) 9.07z (s, 0.1H), 8.20-7.60 (m, 10H), 7.56-7.33 (m, 4H), 7.20-6.80 (m, 2H). δ (ppm) MS (MALDI) m/z = 448.0, theor. calc. 448.1(M•⁺).

**(rac)-bis(thiadiazole)-[8]helicene.** 2,7-bis(benzothiadiazol)vinyl)naphthalene **2** (90 mg, 0.2 mmol, 1 equiv.) was dissolved in 350 mL of toluene in a photoreactor with a catalytic amount of iodine (ca. 0.1 equiv.) and the mixture was irradiated with an immersion lamp (150 W) for 20 h. After evaporation of the solvent and column chromatography (SiO₂, PE: CH₂Cl₂ 4:6) the compound was obtained as a yellow powder (40 mg, 44%). ¹H NMR (300 MHz, CDCl₃): δ (ppm) 8.17 (d, J = 8.2 Hz, 2H), 8.08 (s, 4H), 7.74 (d, J = 8.2 Hz, 2H), 7.51 (d, J = 9.2 Hz, 2H), 7.38 (d, J = 9.2 Hz, 2H). ¹³C NMR (76 MHz, CDCl₃) δ (ppm) 153.76, 151.80, 133.25, 132.38, 132.18, 131.79, 129.81, 129.53, 128.54, 128.44, 127.84, 127.59, 126.17, 125.47, 123.66, 118.96. MS (MALDI) m/z = 444.0, theor. calc. 444.0 (M•⁺).

**Sample preparation.** Monolayer $NiO_x$ and $NiFeO_x$ islands were synthesized by liquid exfoliation from the bulk counterparts reported elsewhere[22,70] and spin-coated on Au(111) film on mica substrates (epitaxial gold on mica, Georg Albert PVD – Beschichtungen). The drop-casting method was used for the samples shown in Fig. 3c, d. The drop-casting method was used for the samples shown in Fig. 3c, d, with 80 μL and 20 μL ink (1 mg $NiO_x$ per 15 mL), respectively, to obtain the same catalyst loading for the comparisons of $NiO_x$ on Au and the sandwich catalyst-molecule-Au configuration. $Ni_9FeO_x$ on Ni foam electrodes were synthesized using a combustion method[44].

All molecules were first dissolved in dichloromethane (DCM, Sigma-Aldrich, puriss. p.a., ≥99.9% (GC)). The molecule solutions were drop-casted on the electrode surfaces to deposit a quasi-monolayer of molecules. After the DCM evaporated completely and all the molecules landed on the surfaces, the electrodes were rinsed with excessive DCM to remove molecules not adsorbed directly on the electrode surface. Afterwards, the electrodes were rinsed with Milli-Q water and ready for the OER activity measurement with adsorbed molecules.

The catalytic activity of Au surfaces is highly dependent on the surface morphology[71]. Under OER conditions, surface reconstruction continuously occurs on Au surfaces and strongly affects the OER activity[40]. Therefore, OER activity measurements of helicene molecule-functionalized Au surface used freshly prepared Au substrates (Au(111) film on mica substrates). Au samples for bare Au experiments were treated using pure DCM following the helicene deposition procedure; however, without helicene molecules dissolved in DCM. This treatment is to rule out the effect of the deposition procedure on other surface properties (e.g., roughness) that may influence the activity comparison. Identical experiments (activity and Pb stripping measurements) were conducted on bare Au and helicene molecule-functionalized Au.

**Electrochemical characterization.** The electrochemical measurements were conducted in a three-electrode cell. The overall activity depends on the absolute amount of catalysts deposited on the Au surface, which varies slightly between samples. An unpurified KOH solution (1 N solution in water, ACROS Organics™) has been used in this work. The measurements were kept running until stable CVs were observed. All electrochemical measurements were conducted using a VSP-300 (BioLogic) potentiostat. A coiled Au wire (99.9%, Alfa Aesar) was used as the counter electrode (CE). The CE was cleaned by flame annealing and rinsed with Milli-Q (Millipore) water before each set of measurements. A HydroFlex® standard hydrogen reference electrode (Gaskatel) was used as the reference electrode (RE). The activity measurements were done without iR-correction to avoid over-compensation and only compare the effect of molecule deposition to the overall catalytic performance.

The effect of DCM washing on the OER activity is shown in Supplementary Fig. 3. The activity was measured before and after washing the electrode with pure DCM. Afterwards, (P)-thiadiazole-[7]helicene molecules were deposited, and activity was measured again. The pure DCM washing had no noticeable effect on the activity, and enhancement can only be observed after (P)-thiadiazole-[7] helicene deposition. In this case, a Ag/AgCl (3 M NaCl) RE (BASi®) was used regarding the concern of using standard hydrogen reference electrodes for OER experiments in alkaline media from Garcia et al.[72]. The working electrode (WE) potential was converted to the RHE scale. Hence, the use of a Ag/AgCl RE likely leads to the same results as using the HydroFlex® standard hydrogen reference electrode in evaluating the chiral molecular functionalization effect. However, due to the lack of stability of the Ag/AgCl RE in long-term experiments[72], the HydroFlex® standard hydrogen reference electrode was used for the rest of the experiments reported in this work.

The EIS measurements were conducted as reported by Watzele et al.[34]. For $NiO_x$ electrodes, the spectra were measured at the potential of 1.6 V vs. RHE (1.59 V vs. RHE in the case of $NiFeO_x$) close to the "onset" potential of the OER where reaction intermediates cover all active sites at $NiO_x$ without interruptions from $O_2$ bubble formation. Each spectrum contains data at 198 frequencies from 30 kHz to 10 Hz measured using a probing signal amplitude of 10 mV, and the data fitting was done using EIS Data Analysis 1.3 software[73,74].

Chronoamperometry measurements were conducted in 0.25 M $CO_2$ saturated $K_2CO_3$ (prepared from Milli-Q water and Potassium carbonate powder, anhydrous, 99.99% trace metals basis, Sigma-Aldrich) to reduce the decomposition of $H_2O_2$. Each chronoamperometry measurement lasted 30 min at 2.5 V vs. RHE to have sufficient production of $H_2O_2$[46,49]. Afterwards, the electrolytes were taken into cleaned polypropylene centrifuge tubes and shaken well for $H_2O_2$ determination.

**H$_2$O$_2$ determination**. The H$_2$O$_2$ determination procedure was introduced by Fuku et al.[75]. 500 µL of each electrolyte was added into 450 µL of 1 M HCl (prepared from Milli-Q water and hydrochloric acid semiconductor grade PURANAL™, fuming 37%, 37–38%, Sigma-Aldrich). Afterwards, 50 µL of 0.1 M FeCl$_2$ (from Iron(II) chloride tetrahydrate, puriss. p.a., ≥99.0% (RT), Sigma-Aldrich) in 1 M HCl was added. The well-mixed liquids were then taken to UV-visible spectro-photometry measurements using A Varian Cary 500 spectrophotometer. The peak around 335 nm appeared due to the change of Fe$^{2+}$ to Fe$^{3+}$ caused by the H$_2$O$_2$ in the electrolyte.

**STM and AFM measurements**. STM and AFM have been used to measure the deposition of the helicene molecules. A Bruker MS10 EC-STM with a NanoScope® V controller controlled by the software Nanoscope 8.15 has been used for the STM measurements. NCHR tips (NanoWorld®) were used for the AFM measurements in Tapping Mode with a Bruker Dimension FastScan AFM. Image analysis was done using the WSxM v5.0 Develop 9.1 software[76] and molecular-scale models using LMAPper and Avogadro software. In the case of STM, nonanoic acid was added on the sample surface as the solvent for a solid/liquid interface to obtain a greater resolution than the one achieved at the solid/air interface[77,78].

## Data availability

Source data are provided as a Source Data file. Further data that support the findings of this study is available from the corresponding authors upon request. Source data are provided with this paper.

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

## Acknowledgements

We heartily thank Prof. Jean-Philippe Ansermet for many helpful discussions and Prof. Hugo Dil for sharing his input on the discussion of the CISS effect in terms of spin polarization. S.L. acknowledges financial support by the Marie Skłodowska-Curie Fellowship (No. 838367 to S.L.) under the European Union's Horizon 2020 research. K.M., N.Z., N.V., and N.A. acknowledge the support from the CNRS, the University of Angers, and the RFI LUMOMAT (grant to K.M.) in France. M.L. and N.A. thank the Swiss Academies of Arts and Sciences (SATW) and the French Ministry of Foreign Affairs for financial support through the bilateral Germaine de Staël (PHC Project 47931VB) project. J.R.G.M. and F.A.G.P. thank the financial support of the FEDER/Ministerio de Ciencia e Innovación, Agencia Estatal de Investigación (RTI2018-095618-B-I00); the Generalitat de Catalunya (2017-SGR-1406), the CERCA Programme/Generalitat de Catalunya.

## Author contributions

Y.L., K.B., N.A., and M.L. conceived and designed the experiments. N.A. and M.L. supervised the project and led the collaboration efforts. Y.L. and K.B. prepared the hybrid electrodes and carried out the scanning probe and electrochemical measurements. K.M., N.Z., and N.V. synthesized the helicenes and performed the enantiomer resolution. S.L., X.H., F.A.G.P., and J.R.G.M. synthesized the catalysts. All authors participated in the result discussions. The manuscript was written by Y.L., N.A., and M.L. with contributions from all the authors.

## Competing interests

The authors declare no competing interests.
