## [Peer Review File · Nature Communications]

Enhancement of Electrocatalytic Oxygen Evolution by Chiral Molecular Functionalization of Hybrid 2D ElectrodesREVIEWER COMMENTS

Reviewer #1 (Remarks to the Author):

Review document attached –

Title: Enhancement of electrocatalytic oxygen evolution by chiral molecular functionalization of hybrid 2D electrodes

Authors: Y. Liang, ... M. Lingenfelder

Ms. No.: NCOMMS-21-44972-T

The authors have modified a NiOx/Au electrode used for the oxygen evolution reaction by adsorbing chiral molecules (functionalized heptahelicenes). As a consequence, they observed enhanced OER activity that they attribute to spin polarization of electrons transported from the Au surface, through the chiral organic monolayer to the NiOx. They demonstrate that the enhancement is not dependent on the handedness of the chiral modifier but that it does depend on the nature of the modifier. The heptahelicenes enhance OER current by ~80% while the use of a functionalized octahelicene yields an enhancement of 131.5%.

This manuscript reports some important and interesting results. However, there are aspects of the work that need further thought before it should be considered for publication.

1. It is not clear to me how accurately the OER enhancement can be measured. The OER current on a bare NiOx/Au electrode is reported five times in Figures 1a, 1b, 2, 3a, and 3b. At a potential of 1.65 V the currents measured on these nominally identical electrodes are: 0.29, 0.41, 2.8, 0.19, and 0.08 mA. Given the large variance in the reported values on the unmodified electrodes, how are the enhancements being reported meaningfully for the chirally modified electrodes?
2. Related to the above, the data for the OER using the octahelicene do not warrant reporting the enhancement (131.5%) with four digits of precision. What would the variance be for five measurements on the same electrode or more importantly on five nominally identical electrodes
3. Related to point two, what is the maximum possible OER enhancement that one could expect because of spin polarization? My naïve guess is that it should be no more than a factor of 2× or 100%. Perfect spin polarization should double the current of electrons with the preferred spin orientation.
4. Figure 4 illustrate the helicenes sandwiched between the NiOx particles and the Au. Is there any direct evidence for this configuration?
Lesser points.
5. p. 3, par. 1, ln. 7. What is the meaning of 'correlate the binding energy of the variables on one adsorbate...'
6. Why is the current density reported in Figure 2, but the absolute current is used in Figures 1 and 3?
7. In Figure 6 the authors mentioned that there is a difference in the coverages of the heptahelicene and the octahelicene. The STM images should allow quantification of this statement.

Reviewer #2 (Remarks to the Author):

This manuscript describes how coating of an electrodes by hetero-helicenes boosts the oxygen evolution reaction (OER) by more than 100% relative to a state-of-the-art catalysts. The enhancement is related to the spin-polarization of the transferred electrons by the chiral molecules. The results presented are very convincing and merit publication. However, the authors should relate to the following comments:

1. Stability of the helicene coated electrodes- How the electrodes behave as a function of time, pH etc.
 2. Did the authors monitored the formation of hydrogen per-oxide? It was reported before that if the electrons' spin is controlled the production of the hydrogen peroxide is reduced. This is an excellent indication for the role of the spin in the enhancement.
 3. There is a former study that indicates the spin filtering properties. It should be mentioned. [Adv. Mat. 28, 1957 (2016); J. Phys. Chem. Lett. 2018, 9, 2025].
 4. The authors conclude that the helicene are spin polarizers and not spin filters. This is not consistent with former studies on the CISS effect. See for example: Phys. Chem. Chem. Phys. 2019, 21, 3761; JPC C 124, 10776 (2020)].
- The reason that the current is not reduced with chiral molecule is their higher conduction for the "correct" spin as demonstrated in: JPC C 124, 10776 (2020). In my opinion this statement should be removed.

Reviewer #3 (Remarks to the Author):

This work explores the chirality-induced-spin-selectivity (CISS) effect on the oxygen evolution reaction (OER) using the nickel-based electrocatalysts. The study is interesting in such, especially since the role of spin-polarization effects have not been extensively investigated for catalytic reactions. The authors have chosen the nickel and the nickel-iron catalyst, which are one of the most interesting catalysts for OER. The manuscript is well written overall, although, a few details are missing, and I suggest the authors to run a few control experiments to prove the chirality effect beyond doubt. Therefore, I suggest major revisions before consideration of this manuscript. Please see my specific comments below.

- 1)) In the abstract, it will be good if the authors are more specific and mention which catalysts are investigated in this study (i.e., Ni and NiFe). I think the abstract is a bit too general right now.
- 2) The authors claim it is better not to purify the electrolyte from Fe-impurities to have more reproducible results, although, I do not agree here. Without electrolyte purification, how do you make sure that electrolyte Fe impurities are not responsible for the enhancement effects? I suggest that the authors consider the additional article from Boettcher's group on Fe impurities and surface effects (<http://dx.doi.org/10.1021/jacs.7b07117>), and the recent work from Markovic's group on Fe-impurity effects difficult to detect without purification (<https://doi.org/10.1021/jacs.0c08959>). Since Fe cations bound to the surface of oxidized Au also have been shown to enhance the OER activity (<http://dx.doi.org/10.1002/celc.201500364>), control experiments of the bare Au substrate are important in absence/presence of Fe impurities. I strongly suggest that the authors repeat some of the experiments in purified KOH and ideally eliminating all glassware from the experiments (either the data in either Figure 1 or Figure 2, for example). Only that would prove the chirality effect beyond doubt.
- 3) When looking at the OER activity data in Figure 1-3, I am immediately wondering what cycle number is shown here? Also, how does the OER activity change during the first 20 cycles for the Ni and NiFe catalysts, respectively? Is there an activation behaviour? Please provide more information.

- 4) What is the typical metal loading on your electrode? Did you determine this by elemental analysis, or how did you control the loading? Also, how is the coverage on the electrode? Does the catalyst film cover the entire Au substrate? Please provide more information and reasoning.
- 5) Was the electrolyte sparged of O₂ before the measurements? I am wondering since you used an RHE reference electrode. Having dissolved O₂ in the electrolyte may induce a potential shift in your RHE. Please consider the following reference on the topic by Koper's group (<https://doi.org/10.1021/acscatal.8b01447>).
- 6) The information on the EIS fitting in Supplementary Fig. S1 is a bit scarce. Please provide the information on the equivalent circuit model, and all assumptions needed to calculate the surface area of the Ni and NiFe catalysts. Please provide references on the topic as well. How did you make sure that the surface area originates only from the catalyst film? Please provide control measurements of the bare Au substrate with and without the chiral molecule immobilized, and compare this to the catalyst. Also, at which potential did you carry out the EIS, open circuit? Does the EIS change during applied OER potentials? Please make sure that you provide these details.
- 7) This relates to the previous question. Since you state that your NiO_x catalyst is an exfoliated 2D material (i.e., a monolayer), the integrated redox peak area (i.e., the peak-charge, Q) should be directly proportional to the surface area (i.e., all Ni atoms should be electrolyte accessible). My question to you is, how do these two methods compare, i.e., if you normalize your data to either the ESCA obtained by EIS or to the moles of Ni atoms on your electrode obtained from integration of the Ni²⁺/Ni^{3+/4+} redox peak? My point is that I want to make sure that you do not distort the OER activity trends by your selected normalization method, especially since metal loadings were never double-checked by elemental analysis if I understood correctly. Just keep in mind that Fe impurities/dopants often impact/suppress the Ni²⁺/Ni^{3+/4+} redox peak, so you may not be able to compare Ni and NiFe films straight off (<http://dx.doi.org/10.1021/jacs.6b00332>). (Ideally, I would strongly advise that the authors double check a few electrodes using elemental analysis, to see how this compares with your selected normalization approach.) Please provide a direct comparison of selected raw data (CVs), and the data after normalization to either the EIS/ESCA method or to the number of Ni atoms obtained from the integrated nickel redox-peak.

Point-by-point response to the reviewers' comments:

Reviewer #1 (Remarks to the Author):

The authors have modified a NiO_x/Au electrode used for the oxygen evolution reaction by adsorbing chiral molecules (functionalized heptahelicenes). As a consequence, they observed enhanced OER activity that they attribute to spin polarization of electrons transported from the Au surface, through the chiral organic monolayer to the NiO_x. They demonstrate that the enhancement is not dependent on the handedness of the chiral modifier but that it does depend on the nature of the modifier. The heptahelicenes enhance OER current by ~80% while the use of a functionalized octahelicene yields an enhancement of 131.5%. This manuscript reports some important and interesting results. However, there are aspects of the work that need further thought before it should be considered for publication.

Response: We thank the reviewer for this positive opinion about our work, and we revised the manuscript according to the suggestions.

1. It is not clear to me how accurately the OER enhancement can be measured. The OER current on a bare NiO_x/Au electrode is reported five times in Figures 1a, 1b, 2, 3a, and 3b. At a potential of 1.65 V the currents measured on these nominally identical electrodes are: 0.29, 0.41, 2.8, 0.19, and 0.08 mA. Given the large variance in the reported values on the unmodified electrodes, how are the enhancements being reported meaningfully for the chirally modified electrodes?

Response: The NiO_x/Au electrodes consist of monolayer NiO_x islands on Au substrates. Our goal is to show the effect of chiral molecular functionalization on the electrocatalytic activity. Therefore, we deposited a submonolayer coverage of 2D NiO_x on Au. The effect from the evolution of electrochemically active surface area (ECSA) during the reaction should be avoided. The NiO_x islands should be spread very well all over the Au surface, and overlapping of islands should be minimized. Therefore, the spin-coating method has been used. However, the amount of deposited NiO_x is difficult to maintain at a constant value. We would like to remind the Reviewer that Figure 2 shows the current density normalized by the ECSA obtained by EIS measurement, not the absolute current. Hence, the value, 2.8, is much higher than the others. Moreover, we only compare the activity (current) change of the same electrode before and after chiral molecular functionalization to prevent any influence from the different loading of NiO_x islands of different electrodes.

2. Related to the above, the data for the OER using the octahelicene do not warrant reporting the enhancement (131.5%) with four digits of precision. What would the variance be for five measurements on the same electrode or more importantly on five nominally identical electrodes.

Response: We thank the reviewer for the comments. We add the statistical summary from 5 nominally identical electrodes in **Supplementary Figure 5** to show the consistency of the enhancement, and we have changed the enhancement factors to have two digits of precision.

Action taken: Statistical summary of thiadiazole-[7]helicene enantiomers and (*M*)-bis(thiadiazole)-[8]helicene functionalization effect respectively on the OER activity of five nominally identical electrodes has been added as **Supplementary Figure 5**.

3. Related to point two, what is the maximum possible OER enhancement that one could expect because of spin polarization? My naïve guess is that it should be no more than a factor of 2x or 100%. Perfect spin polarization should double the current of electrons with the preferred spin orientation.

Response: We appreciate the reviewer's interpretation that the maximum enhancement should be no more than 100 % in the case of perfect spin polarization. The helicene molecules polarize the electron spin. However, in our opinion, the OER activity of a catalyst is not simply determined by the electron spin direction. Spin polarization affects OER kinetics in multiple ways. It changes the binding energy of O species on the catalyst surface, and it is expected that the activation energy should also be affected. In this work, we report the values we obtain from our well-controlled experiments using state-of-the-art materials. A considerable amount of NiO_x sites are not affected by the chiral molecules due to the catalyst-molecule configuration. The limited amount of active sites boosted by the chiral molecular functionalization resulted in more than 100% enhancement in the case of bis(thiadiazole)-[8]helicene. We can conclude that the enhancement factor at a single active site is even higher. We anticipate that higher enhancements can be reached with the ideal catalyst-molecule configuration and optimized chiral molecules suggested in the manuscript, as shown in newly-added **Fig. 3 c**, where helicene molecules are in between the catalysts and the Au substrate. Moreover, although it is not in the scope of this work, the chiral molecular functionalization effect on the mass transfer processes can be envisaged to be further investigated.

4. Figure 4 illustrate the helicenes sandwiched between the NiO_x particles and the Au. Is there any direct evidence for this configuration?

Response: The illustration in the original Figure 4 of the catalyst-molecule-substrate configuration was a conclusion from our measurements and literature, a guideline for the optimized configuration for practical applications. In the current version we have added the OER activity of a sandwiched configuration using (*M*)-bis(thiadiazole)-[8]helicene as **Fig. 3 c**. This is achieved by depositing first the molecule and later the catalyst.

A similar enhancement trend is obtained when depositing first the catalyst and then the molecules. The reason is that the molecules do not attach to the NiO_x but to the Au surface. It is important to note that the molecular layer is not static. During the OER, the molecules diffuse underneath the NiO_x islands. By adding the molecules to the sample after catalyst deposition and OER and then running OER again on the same samples, we can study the sole effect of chiral molecular functionalization. Moreover, this experiment procedure prevents the influence of different catalyst loadings/structures and different resistances of different samples on the OER activity.

Lesser points.

5. p. 3, par. 1, ln. 7. What is the meaning of 'correlate the binding energy of the variables on one adsorbate...'

Response: This sentence, “the scaling relationships correlate the binding energy of the variables of one adsorbate species to a catalyst surface”, is a short expression of the scaling relationships. In the case of the OER, oxygen adsorbate variables include HO^* , HOO^* and O^* . The binding energies of the variables adsorbed to the catalyst are linearly dependent on each other. While changing the binding energy of one of the variables on the catalyst surface (e.g., by changing the catalyst material), the binding energy of another variable also changes linearly.

Action taken: This sentence has been replaced by a more clear one: Moreover, the binding energies of the variables of one adsorbate species to a catalyst surface follow a scaling relationship with each other. For instance, the adsorption free energies of OER reaction intermediates OH^* and OOH^* adhere to a simple relation, $\Delta G_{\text{OOH}^*} = \Delta G_{\text{OH}^*} + 3.2 \pm 0.2 \text{ eV}$, for a large number of catalysts.

Why is the current density reported in Figure 2, but the absolute current is used in Figures 1 and 3?

Response: Our work aims to demonstrate the chiral molecular functionalization effect and its potential to boost the electrocatalytic OER beyond the volcano limits. Therefore, 2D structured electrodes, i.e., 2D NiO_x islands spin-coated on Au(111) surfaces, are used. We attempted to minimize all other factors, for instance, changes in the ECSA and mass transfer issues (e.g., gas bubble formation). Therefore, we mainly report the changes in the overall current. Chiral molecular functionalization effect on the current density of 2D NiO_x islands is shown in the new **Supplementary Figure 1** as a further confirmation.

In Figure 2, we compare the changes in the activity of the 2D structured model electrodes and NiFe-based catalysts deposited on Ni foam (potentially used in practical applications). The former has 2D catalyst islands scattered on the Au surface, and the latter has a 3D structure with the ECSA much larger than the geometric surface area. Therefore, current density should be used to compare the chiral molecular functionalization and the Fe-doping effects quantitatively.

6. In Figure 6 the authors mentioned that there is a difference in the coverages of the heptahelicene and the octahelicene. The STM images should allow quantification of this statement

Response: We thank the reviewer for this suggestion. A quantitative analysis of the number of molecules on Au surface shows the (*M*)-bis(thiadiazole)-[8]helicene molecules were more closely packed (ca. 1.1 molecules/ nm^2) than the (*P*)-thiadiazole-[7]helicene molecules (ca. 0.75 molecules/ nm^2).

Action taken: we have added the number of molecules/ nm^2 .

Reviewer #2 (Remarks to the Author):

This manuscript describes how coating of an electrodes by hetero-helicenes boosts the oxygen evolution reaction (OER) by more than 100% relative to a state-of-the-art catalysts. The enhancement is related to the spin-polarization of the transferred electrons by the chiral molecules. The results presented are very convincing and merit publication. However, the authors should relate to the following comments:

Response: We highly appreciate the reviewer's positive opinion on our work, and we provided point-to-point responses to the comments below.

1. Stability of the helicene coated electrodes- How the electrodes behave as a function of time, pH etc.

Response: We thank the reviewer for mentioning the stability of the helicene-coated electrodes. Stability is one of our top concerns. We selected the helicene with the thiadiazole functional group to have a rigid molecular structure and a stable bond to the electrode. As we stated in the manuscript: "The measurements were kept running until stable CVs were observed", each OER measurement lasted at least 50 min (≥ 16 CV cycles at 5 mV/s). Moreover, we have conducted EIS measurements to determine the ECSA and repeated the OER measurements after EIS to confirm the enhancement. We concluded that the enhancement effect is stable over time.

The pH effect is not in the scope of this work, as alkaline water splitting at Ni-based catalysts is one of the most well-studied electrocatalytic systems. We have focused on the effect of chiral molecular functionalization to further boost the OER reaction. pH dependency is being studied in other systems and is beyond the scope of the present communication. However, we would like to point out that 2,1,3-benzothiadiazole derivatives are extremely stable in acidic and basic media (see for example Weinstock, L. M., & Pollak, P. I. (1968). *The 1,2,5-Thiadiazoles. Advances in Heterocyclic Chemistry*, 107–163.), therefore the thiadiazole-helicene molecules are stable in the experimental conditions we employed here.

2. Did the authors monitored the formation of hydrogen per-oxide? It was reported before that if the electrons' spin is controlled the production of the hydrogen peroxide is reduced. This is an excellent indication for the role of the spin in the enhancement.

Response: We appreciate the reviewer's suggestion. We have conducted experiments to quantitatively determine the chiral molecular effect on the selectivity of oxygen evolution reaction as described in [*ACS Energy Lett.* 2018, 3, 10, 2308–2313; *ChemistrySelect* 2016, 1, 5721–5726]. The H₂O₂ production at NiO_x is significantly reduced by (*M*)-bis(thiadiazole)-[8]helicene.

Action taken: We have conducted experiments to quantitatively determine the chiral molecular effect on the selectivity of H₂O₂ production, and the results have been added as **Fig. 3 d** and **Supplementary Figure 6**.

3. There is a former study that indicates the spin filtering properties. It should be mentioned. [*Adv. Mat.* 28, 1957 (2016); *J. Phys. Chem. Lett.* 2018, 9, 2025].

Response: We thank the reviewer's suggestions and added these references in the revised manuscript.

Action taken: References have been added in the manuscript and highlighted in yellow.

4. The authors conclude that the helicene are spin polarizers and not spin filters. This is not consistent with former studies on the CISS effect. See for example: Phys. Chem. Chem. Phys. 2019, 21, 3761; JPC C 124, 10776 (2020)]. The reason that the current is not reduced with chiral molecule is their higher conduction for the “correct” spin as demonstrated in: JPC C 124, 10776 (2020). In my opinion this statement should be removed.

Response: We thank the reviewer for the remark. Our results agree with the current understanding of the CISS effect as a result of spin exchange interaction, where adsorption of chiral molecules spin-polarizes the interface. We have adapted the statement in the manuscript and cited JPC C 124, 10776 (2020).

Reviewer #3 (Remarks to the Author):

This work explores the chirality-induced-spin-selectivity (CISS) effect on the oxygen evolution reaction (OER) using the nickel-based electrocatalysts. The study is interesting in such, especially since the role of spin-polarization effects have not been extensively investigated for catalytic reactions. The authors have chosen the nickel and the nickel-iron catalyst, which are one of the most interesting catalysts for OER. The manuscript is well written overall, although, a few details are missing, and I suggest the authors to run a few control experiments to prove the chirality effect beyond doubt. Therefore, I suggest major revisions before consideration of this manuscript. Please see my specific comments below.

Response: We thank the reviewer for the positive opinion on our work and all the constructive suggestions. We have conducted additional experiments and revised the manuscript accordingly.

- 1) In the abstract, it will be good if the authors are more specific and mention which catalysts are investigated in this study (i.e., Ni and NiFe). I think the abstract is a bit too general right now.

Response: We appreciate this suggestion and modified the abstract.

Action taken: The name of catalysts and chiral modifiers used in this study have been added in the **abstract**.

- 2) The authors claim it is better not to purify the electrolyte from Fe-impurities to have more reproducible results, although, I do not agree here. Without electrolyte purification, how do you make sure that electrolyte Fe impurities are not responsible for the enhancement effects? I suggest that the authors consider the additional article from Boettcher's group on Fe impurities and surface effects (<http://dx.doi.org/10.1021/jacs.7b07117>), and the recent work from Markovic's group on Fe-impurity effects difficult to detect without purification (<https://doi.org/10.1021/jacs.0c08959>). Since Fe cations bound to the surface of oxidized Au also have been shown to enhance the OER activity (<http://dx.doi.org/10.1002/celec.201500364>), control experiments of the bare Au substrate are important in absence/presence of Fe impurities. I strongly suggest that the authors repeat some of the experiments in purified KOH and ideally eliminating all glassware from the experiments (either the data in either Figure 1 or Figure 2, for example). Only that would prove the chirality effect beyond doubt.

Response: We agree with the reviewer that electrolyte Fe impurities introduce an inevitable enhancement effect on the activity of Ni-based catalysts. We were aware of the articles about Fe impurities from Boettcher's and Markovic's groups. As stated in the Methods section, we kept running the OER activity measurement of fresh NiO_x until a stable CV was observed. We often conducted EIS measurements to determine the ECSA alongside the activity measurements and then repeated the activity measurements to ensure stable activity results. We think catalysts had already been fully affected by the Fe impurities before depositing helicene molecules. Fe impurities should not induce further change in the activity after molecule deposition.

We value the reviewer's comment. To give further evidence, we have repeated the measurements shown in **Fig. 1 a** in purified KOH (purification procedure as used in Boettcher's article J. Am. Chem. Soc. 2014, 136, 6744–53). The result has been added in modified **Fig. 2 a** as another example of the compatibility of chiral molecular functionalization with the Fe-doping effect.

This work aims to demonstrate the chiral molecular functionalization effect on state-of-the-art OER catalysts. Our priority is to reveal the activity change solely caused by the chiral molecule deposition. As cited in the manuscript, Farhat et al. reported that NiO_x in unpurified KOH, compared with NiO_x in purified KOH and NiFeO_x in both purified and unpurified KOH, showed the most stable activity over time. Therefore, we chose to conduct most of the OER measurements in unpurified KOH.

Action taken: OER activity measurement of NiO_x on Au before and after (*P*)-thiadiazole-[7]helicene deposited in purified (Fe-free) KOH in a cell without glass parts has been added in the modified **Fig. 2 a**.

3) When looking at the OER activity data in Figure 1-3, I am immediately wondering what cycle number is shown here? Also, how does the OER activity change during the first 20 cycles for the Ni and NiFe catalysts, respectively? Is there an activation behaviour? Please provide more information.

Response: We agree with the reviewer that there is activation behavior, especially on freshly prepared electrodes. However, because we ran the OER over an excessive time to achieve a stable CV before recording the data shown in the manuscript, the catalyst should be fully "activated". The CVs recorded from freshly prepared samples in Fe-free and unpurified KOH have been added in the new **Supplementary Figure 2**. An increase in the activity can be seen in both cases during the first cycles (e.g., activation behavior). The activity is likely more stable in unpurified KOH.

As mentioned above, we first ran the OER CV on a fresh sample until a stable curve was observed. Afterwards, we started recording the data containing the CVs shown in the manuscript. The total cycle number of each measurement is typically between 10-20. We selected the cycles with the same cycle number from the CVs recorded before and after helicene molecule deposition. We still occasionally observed slight decreases in the activity over time, likely due to the catalysts detachment and O₂ bubble formation at the electrode surface. However, the enhancement caused by the helicene is much more significant.

Action taken: We have modified the Electrochemical characterization in the **Methods** session to be more clear on the OER activity measurement. Additionally, the CVs recorded from freshly prepared samples in Fe-free and unpurified KOH have been added in the new **Supplementary Figure 2**.

4) What is the typical metal loading on your electrode? Did you determine this by elemental analysis, or how did you control the loading? Also, how is the coverage on the electrode? Does the catalyst film cover the entire Au substrate? Please provide more information and reasoning.

Response: The ECSA determined by the EIS analysis in **Supplementary Figure 1** is ca. 0.046 cm². The geometric surface area of the electrode is ca. 0.5 cm². The coverage of the NiO_x on the electrode for Fig. 1 a is approximately 9.2%, assuming all the NiO_x surface is

active towards the OER. We did not carry out EIS measurement on all electrodes to minimize the complexity of the experiments and prevent any additional effects or surface area changes to the catalysts and thus concentrated on the molecule deposition effect.

This work aims to demonstrate the chiral molecular functionalization effect on the enhancement of OER activity. We designed the experimental procedure to ensure the enhancement is solely due to the helicene molecule deposition. Therefore, deposition of 2D NiO_x islands was achieved by spin coating methods to best spread the islands and prevent any clustering of the islands. The Au substrate is not fully covered by the islands, and the spare areas are for stable molecule adsorption.

We did not conduct elemental analysis and did not precisely control the metal loading. As mentioned above, we do not compare the activity of different electrodes in general, even nominally identical electrodes. We first measured the stable activity of one electrode and then deposited the molecules on the same electrode. After the removal of excessive molecules, we carried out the same activity measurement (i.e., CV) until the CV was stable. Each activity comparison reported in this manuscript is taken from the same electrode, except the sandwiched configuration shown in **Fig. 3 c**.

- 5) Was the electrolyte sparged of O₂ before the measurements? I am wondering since you used an RHE reference electrode. Having dissolved O₂ in the electrolyte may induce a potential shift in your RHE. Please consider the following reference on the topic by Koper's group (<https://doi.org/10.1021/acscatal.8b01447>).

Response: We thank the reviewer for sharing with us the concern of the stability of the reference potential. We conducted all OER activity measurements in O₂-saturated KOH, and only stable CVs were taken into account. We have also conducted the experiment using a Ag/AgCl reference. We have added this result as the new **Supplementary Figure 3**.

Action taken: (*P*)-thiadiazole-[7]helicene effect on the OER activity of NiO_x using a Ag/AgCl reference has been added as the new **Supplementary Figure 3**. The numbering of the Supplementary Figures has been changed accordingly. A description of this experiment has been added in the **Methods** session as part of the **Electrochemical characterization**. The results also show that rinsing the sample with pure DCM has no effect on the OER activity.

- 6) The information on the EIS fitting in Supplementary Fig. S1 is a bit scarce. Please provide the information on the equivalent circuit model, and all assumptions needed to calculate the surface area of the Ni and NiFe catalysts. Please provide references on the topic as well. How did you make sure that the surface area originates only from the catalyst film? Please provide control measurements of the bare Au substrate with and without the chiral molecule immobilized, and compare this to the catalyst. Also, at which potential did you carry out the EIS, open circuit? Does the EIS change during applied OER potentials? Please make sure that you provide these details.

Response: The EIS-based methodology used for the determination of the ECSA of NiO_x is from a recent publication (Ref. 30, ACS Catal. 2019, 9, 10, 9222–9230). We agree with the reviewer that this methodology is a bit scarce for some audiences. The EIS data was taken at the potential close to the onset potential (i.e., at low overpotentials) of OER at NiO_x, and the adsorption capacitance was used to determine the ECSA. This methodology has been proven and used in a number of other studies. At this particular potential, 1.6 V vs. RHE in the case

of NiO_x (1.59 V vs. RHE in the case NiFeO_x), only reaction intermediates (OH*, *OOH, and O*) fully adsorbed on the active sites of the catalyst contribute to the adsorption capacitance. We have attempted to perform control measurements on bare Au. However, we did not obtain meaningful EIS data. Without the specific adsorption of the oxygen species on Au at 1.6 V vs. RHE, the EIS data cannot be fitted to the same equivalent circuit model.

ECSA determination of transition metal oxide catalysts, especially for nanostructured 2D materials, is of great importance for OER activity studies. There are a few methods commonly used, for instance, Brunauer–Emmett–Teller (BET) method and double layer capacitance. We had the same concern as the reviewer that these methods often cannot differentiate between the active and non-active parts of the electrode or take the contribution of the bulk of the metal oxides. Therefore, we decided to use the method reported in ACS Catal. 2019, 9, 10, 9222–9230.

Action taken: We added a brief discussion of the EIS methodology in the **Methods** section.

7) This relates to the previous question. Since you state that your NiO_x catalyst is an exfoliated 2D material (i.e., a monolayer), the integrated redox peak area (i.e., the peak-charge, Q) should be directly proportional to the surface area (i.e., all Ni atoms should be electrolyte accessible). My question to you is, how do these two methods compare, i.e., if you normalize your data to either the ECSA obtained by EIS or to the moles of Ni atoms on your electrode obtained from integration of the Ni²⁺/Ni^{3+/4+} redox peak? My point is that I want to make sure that you do not distort the OER activity trends by your selected normalization method, especially since metal loadings were never double-checked by elemental analysis if I understood correctly. Just keep in mind that Fe impurities/dopants often impact/suppress the Ni²⁺/Ni^{3+/4+} redox peak, so you may not be able to compare Ni and NiFe films straight off (<http://dx.doi.org/10.1021/jacs.6b00332>). (Ideally, I would strongly advise that the authors double check a few electrodes using elemental analysis, to see how this compares with your selected normalization approach.) Please provide a direct comparison of selected raw data (CVs), and the data after normalization to either the EIS/ECSA method or to the number of Ni atoms obtained from the integrated nickel redox-peak.

Response: We thank the reviewer for this suggestion. We also think this comparison is helpful to further validate the activity enhancement reported in this work. However, based on Boettcher's articles (e.g., Nano Lett. 2017, 17, 11, 6922–6926 and ACS Appl. Mater. Interfaces 2019, 11, 6, 5590–5594) and our own ongoing studies, structural evolution takes place at 2D transition metal oxide catalysts, and the 2D islands turn into 3D structures under OER conditions. Although the catalyst synthesis and spin coating provide optimal 2D island deposition on Au surface, we can still find a limited number of bulk NiO_x structures by AFM. Structural evolution resulting in 3D structures occurred before the CV measurements reported in our work because we ran the OER experiments until stable CVs were observed. Please see the added **Supplementary Figure 9** for the AFM images of the structural evolution NiO_x islands. The number of Ni atoms from the analysis of the redox peaks is likely much larger than the number of active atoms during the OER. However, we believe the redox peak analysis can further confirm that the chiral molecules are responsible for the activity enhancement.

Action taken: A comparison of the raw data shown in **Fig. 1 a**, the specific current density using the ECSA determined by EIS method and the current normalized by the number of Ni atoms obtained from the integrated Ni redox peak has been added as **Supplementary Figure 1 c**. AFM images of NiO_x on Au surface have been added as **Supplementary Figure 9**.

REVIEWER COMMENTS

Reviewer #1 (Remarks to the Author):

The authors have made substantial modifications to their manuscript. I think that it is ready for publication.

Reviewer #2 (Remarks to the Author):

The authors responded properly to all the comments. The article should be published.

Reviewer #3 (Remarks to the Author):

I think that the authors have improved the manuscript significantly during the revisions. I am convinced regarding the chiral enhancement of the OER activity. However, I still have a minor issue regarding some measurements (H₂O₂ production) that was added during the review stage, which I strongly recommend the authors to fix before recommending publication of this manuscript in Nature Communications. I therefore suggest minor revisions.

Comment 1)

The authors have included new data showing H₂O₂ production in CO₂ purged K₂CO₃ electrolyte (neutral pH), which demonstrates H₂O₂ formation during OER potentials (2.5 V), and that the chiral molecule suppresses H₂O₂ formation. The purpose of these studies is however unclear, and the information regarding the fundamental aspects of H₂O₂ formation is lacking. This confuses the reader, and also results in a poor understanding of these measurements. First, I find these measurements completely irrelevant for the OER process in 0.1 M KOH (alkaline pH), since the standard potential (E⁰) for H₂O₂ is much higher (1.76 V) compared to O₂ (1.23 V). Furthermore, it is known that H₂O₂ production is limited to bicarbonate electrolytes around neutral pH and does not occur during the experimental conditions used in the manuscript for OER investigations (0.1 M KOH, alkaline electrolyte. This is not mentioned. This is supported by the high FE toward O₂ in alkaline electrolytes (close to 99 % excluding the contribution of Ni oxidation)(<http://dx.doi.org/10.1021/jacs.6b12250>). The pH and choice of anions have a big influence on the faradaic efficiency (FE) of H₂O₂, which is also not mentioned. The FE of O₂ should in any case be determined quantitatively in 0.1 M KOH and not in K₂CO₃ before any discussion related to the OER process. Indeed, I find these H₂O₂ measurements useful for demonstrating the chiral effect of the molecule, however, not for anything else since the relevant data is missing. I strongly recommend the authors to clarify the purpose of their H₂O₂ studies, and to mentioned relevant fundamental aspects regarding H₂O₂ formation, and the dependence on the experimental conditions for the FE toward H₂O₂/O₂. I advise reading the following articles (<https://doi.org/10.1038/s41467-017-00585-6>, <https://doi.org/10.1021/acscatal.8b04873>, <https://doi.org/10.1021/acsaem.1c02258>).

Point-by-point response to the reviewers' comments:

Reviewer #1 (Remarks to the Author):

The authors have made substantial modifications to their manuscript. I think that it is ready for publication.

Response: We highly appreciate the reviewer's positive opinion, and we thank the reviewer again for all the helpful suggestions.

Reviewer #2 (Remarks to the Author):

The authors responded properly to all the comments. The article should be published.

Response: We thank the reviewer for the approval of the revised version and once again for all the valuable suggestions.

Reviewer #3 (Remarks to the Author):

I think that the authors have improved the manuscript significantly during the revisions. I am convinced regarding the chiral enhancement of the OER activity. However, I still have a minor issue regarding some measurements (H₂O₂ production) that was added during the review stage, which I strongly recommend the authors to fix before recommending publication of this manuscript in Nature Communications. I therefore suggest minor revisions.

Response: We are delighted that the reviewer has approved our work. We thank the reviewer for bringing up this last concern to improve the clarity of the manuscript.

Comment

1)

The authors have included new data showing H₂O₂ production in CO₂ purged K₂CO₃ electrolyte (neutral pH), which demonstrates H₂O₂ formation during OER potentials (2.5 V), and that the chiral molecule suppresses H₂O₂ formation. The purpose of these studies is however unclear, and the information regarding the fundamental aspects of H₂O₂ formation is lacking. This confuses the reader, and also results in a poor understanding of these measurements. First, I find these measurements completely irrelevant for the OER process in 0.1 M KOH (alkaline pH), since the standard potential (E⁰) for H₂O₂ is much higher (1.76 V) compared to O₂ (1.23 V). Furthermore, it is known that H₂O₂ production is limited to bicarbonate electrolytes around neutral pH and does not occur during the experimental conditions used in the manuscript for OER investigations (0.1 M KOH, alkaline electrolyte. This is not mentioned. This is supported by the high FE toward O₂ in alkaline electrolytes (close to 99 % excluding the contribution of Ni oxidation)(<http://dx.doi.org/10.1021/jacs.6b12250>). The pH and choice of anions have a big influence on the faradaic efficiency (FE) of H₂O₂, which is also not mentioned. The FE of O₂ should in any case be determined quantitatively in 0.1 M KOH and not in K₂CO₃ before any discussion related to the OER process. Indeed, I find these H₂O₂ measurements useful for

demonstrating the chiral effect of the molecule, however, not for anything else since the relevant data is missing. I strongly recommend the authors to clarify the purpose of their H₂O₂ studies, and to mentioned relevant fundamental aspects regarding H₂O₂ formation, and the dependence on the experimental conditions for the FE toward H₂O₂/O₂. I advise reading the following articles (<https://doi.org/10.1038/s41467-017-00585-6>, <https://doi.org/10.1021/acscatal.8b04873>, <https://doi.org/10.1021/acsaem.1c02258>).

Response: We understand the reviewer's concern over the purpose of the H₂O₂ production measurement under a different electrochemical condition. It is expected that the chiral molecular functionalization effect tailors the selectivity of spin-dependent electron transfer processes. The formation of the O–O bond in the OER proceeds via spin conservation to yield the paramagnetic triplet state of molecular oxygen. Thus, spin polarization of the active catalyst surface may induce parallel spin alignment of oxygen atoms during the reaction, improving the efficiency of the process while suppressing other reactions that are initially spin-independent, as H₂O₂ production. Therefore, the production of H₂O₂ during water oxidation is a suitable example to test the effect on selectivity. However, as the reviewer has already noticed, the experimental conditions for the OER at NiO_x are not applicable for semi-quantitatively studying H₂O₂ production. As the reviewer mentioned, NiO_x based catalysts favor the four-electron oxidation reaction pathway and, therefore, the production of O₂ (e.g., close to 99 % excluding the contribution of Ni oxidation). Moreover, H₂O₂ is not stable in alkaline media. Therefore, to clearly reveal the chiral molecular functionalization effect on the production of H₂O₂, we chose a neutral solution containing bicarbonate and the potential range where a sufficient amount of H₂O₂ can be produced for the UV-Vis measurement. This result confirms that the spin-polarization can significantly affect both catalytic activity and selectivity. Future studies focusing on the effect of spin-polarization on the intermediate reaction steps in water oxidation are essential. However, this is not within the scope of this work.

Action taken: we have taken the reviewer's valuable suggestions and added the clarification of the purpose of the H₂O₂ measurements. All the changes are marked in yellow. We have also added the four references suggested by the reviewer (refs 46-49).

The changes are shown here.

1, page 12:

Water oxidation is a suitable example to test the chiral molecular functionalization effect on product selectivity, as H₂O₂ formation is expected to be reduced at spin-polarized surfaces⁴⁵. However, NiO_x is not among the catalysts (e.g., ZnO⁴⁶, SnO₂ and TiO₂⁴⁷) that selectively produce H₂O₂ during water oxidation. The OER at NiO_x based catalysts already dominates in alkaline media⁴⁸, and the production of H₂O₂ is neglectable in these conditions. Moreover, it is reported that a neutral solution containing bicarbonate enhances the faradaic efficiency towards H₂O₂ production⁴⁹, and the neutral pH also hinders the H₂O₂ from decomposition. In order to test for changes in selectivity induced by the chiral molecules, we used the same

electrode configurations in **Fig. 3 c** to confirm that the chiral molecular functionalization suppresses H₂O₂ production at neutral pH. Chronoamperometry (**Supplementary Figure 6**) and UV-visible spectrophotometry measurements (see **Methods**) were conducted in CO₂ saturated K₂CO₃ (neutral pH and containing a constant concentration of HCO₃⁻). The H₂O₂ production was significantly reduced by the chiral molecular functionalization, as shown in **Fig. 3 d**.

2, page 21:

Chronoamperometry measurements were conducted in 0.25 M CO₂ saturated K₂CO₃ (prepared from Milli-Q water and Potassium carbonate powder, anhydrous, 99.99% trace metals basis, Sigma-Aldrich) to reduce the decomposition of H₂O₂. Each chronoamperometry measurement lasted 30 min at 2.5 V vs. RHE to have sufficient production of H₂O₂^{46,49}. Afterwards, the electrolytes were taken into cleaned polypropylene centrifuge tubes and shaken well for H₂O₂ determination.

⁴⁵ Mtangi, W., Kiran, V., Fontanesi, C. & Naaman, R. The role of the electron spin polarization in water splitting. *J. Phys. Chem. Lett.* **6**, 4916–4922 (2015).

⁴⁶ Kelly, S. et al. ZnO as an active and selective catalyst for electrochemical water oxidation to hydrogen peroxide. *ACS Catal.* **9**, 4593–4599 (2019).

⁴⁷ Shi, X., Siahrostami, S., Li, GL. et al. Understanding activity trends in electrochemical water oxidation to form hydrogen peroxide. *Nat. Commun.* **8**, 701 (2017).

⁴⁸ Görllin, M. et al. Tracking catalyst redox states and reaction dynamics in Ni–Fe oxyhydroxide oxygen evolution reaction electrocatalysts: the role of catalyst support and electrolyte pH. *J. Am. Chem. Soc.* **139**, 2070–2082 (2017).

⁴⁹ Gill, T. M., Vallez, L. & Zheng, X. Enhancing Electrochemical Water Oxidation toward H₂O₂ via Carbonaceous Electrolyte Engineering. *ACS Appl. Energy Mater.* **4**, 12429–12435 (2021).

REVIEWERS' COMMENTS

Reviewer #3 (Remarks to the Author):

I think the authors have addressed all the concerns, and I would recommend this manuscript for publication in Nature Communications.

Point-by-point response to the reviewers' comments:

Reviewer #3 (Remarks to the Author):

I think the authors have addressed all the concerns, and I would recommend this manuscript for publication in Nature Communications.

Response: We would like to express our sincere gratitude to the reviewer for supporting our work providing helpful and valuable suggestions.